# A Multi-Scale Approach to Modeling the Interfacial Reaction Kinetics of Lipases with Emphasis on Enzyme Adsorption at Water-Oil Interfaces

**Sherly Rusli** [1],*[iD]**, Janna Grabowski** [1]**, Anja Drews** [2] **and Matthias Kraume** [1] [iD]

[1] Chair of Chemical and Process Engineering, Technical University of Berlin, Ackerstrasse 76, 13355 Berlin, Germany; janna.grabowski@campus.tu-berlin.de (J.G.); matthias.kraume@tu-berlin.de (M.K.)

[2] HTW Berlin, Department of Engineering II, School of Life Science Engineering, Wilhelminenhofstr. 75A, 12459 Berlin, Germany; Anja.Drews@HTW-Berlin.de

\* Correspondence: s.rusli@tu-berlin.de; Tel.: +49-30-314-72622

**Abstract:** The enzymatic hydrolysis of triglycerides with lipases (EC 3.1.1.3.) involves substrates from both water and oil phases, with the enzyme molecules adsorbed at the water-oil (w/o) interface. The reaction rate depends on lipase concentration at the interface and the available interfacial area in the emulsion. In emulsions with large drops, the reaction rate is limited by the surface area. This effect must be taken into account while modelling the reaction. However, determination of the interfacial saturation is not a trivial matter, as enzyme molecules have the tendency to unfold on the interface, and form multi-layer, rendering many enzyme molecules unavailable for the reaction. A multi-scale approach is needed to determine the saturation concentration with specific interfacial area so that it can be extrapolated to droplet swarms. This work explicitly highlights the correlation between interfacial adsorption and reaction kinetics, by integration of the adsorption kinetics into the enzymatic reaction. The rate constants were fitted globally against data from both single droplet and drop swarm experiments. The amount of adsorbed enzymes on the interface was measured in a single drop with a certain surface area, and the enzyme interfacial loading was estimated by Langmuir adsorption isotherm.

**Keywords:** enzymatic hydrolysis; lipases; interfacial kinetics modeling; protein adsorption

## 1. Introduction

Lipases (EC. 3.1.1.3.) have gained popularity in industrial applications as the main catalyst for hydrolyzing lipids. The enzymatic hydrolysis of triglyceride offers a greener alternative to conventional processes. Utilizing lipases as catalyst, the reaction can be done in mild conditions (30 to 60 °C at atmospheric pressure) compared to the commercial fat splitting process, the Colgate Emery, which is performed at high pressure and high temperature (300 °C at 50 bar) [1,2]. In addition to energy saving potential and less purification efforts, the enzymatic process also preserves the quality of fatty acids without exposing the molecules to harsh temperatures. This is particularly important for processing temperature sensitive materials such as ricinoleic acid [3]. Numerical modelling is a valuable tool in developing such processes, to determine the optimum over a wide range of process conditions. For one, to maximise reaction yield while mitigating enzyme inactivation over time. For the development of any process simulations, the availability of reliable kinetic models are an integral part [4].

Lipases are known for their unique interfacial catalysis, where the enzyme molecules are adsorbed at the water-oil (w/o) interface and react with both substrates in the two immiscible liquids (triglyceride in the oil phase and water in the aqueous phase), as seen in Equation (1) [5].

$$TG + 3H_2O \xrightleftharpoons{\text{lipases}} 3FA + Gly. \tag{1}$$

This phenomenon is a combination of both interfacial activation and the interfacial adsorption of the enzyme [6,7]. The importance of interfacial area in the kinetics was recognized since early research from Verger and de Haas [8]. The reaction rate depends on the specific interfacial area where the enzymes adsorb ($a$) [9]. Following this tendency, many modeling efforts put strong emphasis on interfacial area [4,8,10–17], showing that integration of the interfacial area limitation improves the performance of the kinetic models. Most models employed a pre-step before the enzymatic reaction, where the lipases are adsorbed at the interface (Equation (2)). By doing so, it is possible to introduce interfacial limitation to the model by limiting the amount of enzymes adsorbed at the interface ($E^*$). The lipases in the aqueous phase ($E$) are first adsorbed at the available interfacial area ($a$) to become activated enzymes ($E^*$). Only the activated enzyme molecules ($E^*$) react with glyceride ($G$) as substrate, producing the enzyme complex ($EG$) which release the product fatty acid ($FA$). After the reaction, the activated enzymes ($E^*$) canAction.

$$E + a \underset{k_d}{\overset{k_a}{\rightleftharpoons}} E^* \tag{2}$$

$$E^* + G \underset{k_{-1}}{\overset{k_1}{\rightleftharpoons}} EG \underset{k_{-2}}{\overset{k_2}{\rightleftharpoons}} FA + E^*. \tag{3}$$

The reaction kinetics were modelled based on the reversible Michaelis Menten mechanism proposed by Keleti [18]. The textbook Michaelis Menten kinetics introduced by Briggs and Haldane [19] was modelled for irreversible reaction, where there are no influences of the product concentration. This assumption is valid for the beginning of reaction, when the accumulation of product or other intermediates are negligible. However, for modelling the full time course of the reaction, which is needed for implementation in a continuous process, the model must be able to predict not only the initial reaction rate, but also the equilibrium state. Thus the backwards reaction must be included in the model. Nowadays, it is possible to fit the original kinetic constants globally on the basis of numerical integration of the rate equations without simplifying assumptions [20], which leads to the prediction of the final equilibrium state, as demonstrated by numerous research [4,21]. On the downside, the global fitting can lead to arbitrary values of the kinetic constants. The plausibility of these values must be validated against experimental data.

A stirred tank reactor is commonly used to produce the emulsion of water and oil. At this scale, the specific interfacial area ($a$) is calculated from the drop size. Smaller drops result in higher interfacial area, and thus higher reaction rate. Conversely, with larger drops the reaction rate is limited by the low interfacial area. Measurement of drop sizes in an emulsion can be challenging, especially in small reactors. In past research, the drop sizes were experimentally determined ex-situ by analyzing emulsion samples under the microscope [22] or in-situ with laser diffraction [13,16]. The first method can be applied on stable emulsions with very small drops, which means there was no surface limitation. The second method can be applied to emulsions with low dispersed phase volumes. For simplification, many researchers [4,11,15,17] calculated the drop size with empirical correlations using the process parameters of the emulsification process (temperature, power input or energy dissipation rate). In reality, not only the power input or the stirring rate determine the drop size in the emulsion, but also the presence of surface active substances such as enzymes and Mono- and Di-glycerides [23]. These molecules influence the interfacial tension and thus the emulsification process. Heyse et al. [24] reported a decrease of drop sizes while the enzyme concentration is increased, due to the decrease of the interfacial tension.

To model the interfacial limitation, the interfacial area (*a*) must be combined with the enzyme interfacial loading ($\Gamma$). The amount of enzymes adsorbed at the interface significantly limits the initial reaction rate. Most researchers assumed a value of enzyme loading at the interface based on the molecular size of the enzyme [14,25]. This approach assumed the enzymes to be globular or rigid molecules, which occupy a specific area on the interface. Interfacial measurements of protein molecules [5,26,27] showed that large molecules tend to unfold on the interface, leading to larger interfacial occupancy, which in the end leads to an overestimation of the enzyme interfacial loading.

Jurado et al. [13], Albasi et al. [22] determined the saturation of enzyme area coverage from the limitation of the initial reaction rate. This approach assumed abundant free interfacial area for the total enzyme molecules to be adsorbed at. Therefore, the models were only applicable for low enzyme concentrations. At higher enzyme concentrations, multi-layer adsorption of enzymes was observed [28]. Under these conditions, the models overestimated the reaction rate, as the limit of enzymes adsorbed at the interface was not taken into consideration. Al-Zuhair et al. [14] demonstrated that by adapting the apparent adsorption—desorption rate of lipases, it was possible to improve the model for high enzyme concentration.

To simplify the multiphase system, Nury et al. [29] and Labourdenne et al. [30] proposed the use of drop tensiometry for monitoring the hydrolysis of long-chain triacylglycerol. The reaction was performed on a single drop with a certain surface area, at different process conditions (pH, concentration, etc.). This method assumed that the initial decrease of interfacial tension was proportional to the initial interfacial reaction rate, which reflects the lipases activity. Due to the scale, it was not possible to determine the concentration of the reaction species. Since it was not possible to determine the reaction rate, there was no distinct separation between the adsorption process and the reaction itself. However, the experiments highlighted the interfacial activity of the lipases, and, more importantly, the experiments showed that the limiting catalytic rate is correlated with the enzyme concentration on the interface. Using the same drop tensiometry, many researchers [5,28,31–38] focused on the physical adsorption of proteins on liquid-liquid interfaces. The finding supports that many protein molecules, including inert proteins, are surface active and have strong tendency to adsorb at interfaces. It stands to reason that the decrease of interfacial tension from lipases is not solely due to the interfacial reaction, but also due to adsorption.

Despite extensive research on the topic, a gap exists between the two scales, the single drop and the droplet swarm. Enzyme adsorption kinetics must be observed on a specific interfacial area. At the single drop scale, the w/o interfacial area can be controlled, and subsequently the amount of enzyme loaded on the area ($\Gamma$) can be calculated. Although the single drop also acts as reactor on the smallest scale, due to the small volume it is difficult to quantify the product and substrate concentration. Therefore, kinetic modeling was done from experiments in droplet swarms or in an emulsion reactor, where the macro-kinetics can be observed.

This work aims to bridge the different scales by combining the experimental results in one kinetic model, as described in Equation (2) and (3). The rate constants were fitted with multi-scale experiments from a single drop to drop swarms. By doing so, the rate constants were not randomly fitted in a black box, but it is possible to analyze the physical meaning behind them.

The experiments consist of two parts to investigate the effect of lipases at the w/o interface and its effects on the emulsion. The enzyme adsorption process was experimentally monitored in a single drop environment, to determine the enzyme loading on the interface ($\Gamma_\infty$), and the adsorption isotherm ($K_L$). In the second part, hydrolysis reactions were conducted in a stirred tank reactor, against which the reaction rate constants can be fitted. Similar experimental conditions, for example, enzyme concentration in the aqueous phase (*Ce*), temperature, oil purity and so forth, were used in both scales for comparability. The final model is used to determine the optimum condition for the hydrolysis reaction in regards to the interfacial area needed for a certain enzyme concentration. In a stirred tank reactor, this information can be used to optimize the operating stirring speed, which in turn reduces the shear deactivation of the enzyme.

## 2. Materials and Methods

In this section the materials and methodology used in both single drop and drop swarm experiments are listed. Specific experimental setup and analysis in the different scales are listed in the respective sections.

### 2.1. Chemicals

### 2.1.1. Substrates

As source of triglycerides, semi-refined sunflower oil was provided by Oleon GmbH, Emmerich am Rhein, Germany. Sunflower oil mainly consists of 30% oleic acid and 60% linoleic acid, with traces of impurities. Ultra pure water was used as aqueous phase.

### 2.1.2. Biocatalyst

Commercial enzyme Lipomod 034P (L34P) from *Candida rugosa* was provided by Biocatalyst Ltd., Wales, UK, with an activity of 115,000 $Ug^{-1}$ based on olive oil substrate. The powder enzyme was dissolved in pure water as aqueous phase. Lipases from *Candida rugosa* typically has a molecular weight ranging from 30–60 kDa [39,40], which leads to uncertainties in modeling. In this work, the molecular mass of L34P was taken as 40 kDa, based on the work of Duarte et al. [39], which shows that the proteins at this particular molecular mass are mainly responsible for the fat splitting.

### 2.2. Numerical Integration

All parameter fittings were conducted with the Global Optimization Toolbox in MATLAB® 2019b by minimizing the sum of square errors between experimental data and modelling results using fmincon solver. Multivariate solvers GlobalSearch and MultiSearch were used in combination to avoid the local minima.

For adsorption kinetic modelling in Section 3.3, the SSE was calculated between the surface loading ($\Gamma$) calculated from experimental data and from simulation. In Section 4.3, the SSE was calculated for both concentration of product (*FA*) and substrate (*G*).

## 3. Single Drop

A model describing the enzymatic kinetics is necessary for determination of the optimum reaction conditions, in regards to interfacial area limitations (Section 4.3). However, there are several parameters in the system that cannot be measured experimentally. For one, to integrate the interfacial area (*a*) into the model, the loading of the enzyme on the w/o interface ($\Gamma_\infty$) must be defined. This value cannot be measured and therefore needs to be calculated by fitting the time dynamic interfacial tension in a single drop.

### 3.1. Experimental Set-Up and Analysis: The Pendant Drop Method

To gain insight into the enzyme adsorption at the w/o interface, the interfacial tension between the two phases was measured using the pendant drop technique (OCA15EC, DataPhysics GmbH, Filderstadt, Germany), as shown in Figure 1 [41]. A drop of water or enzyme solution was introduced in sunflower oil with a syringe. A camera captured images of the drop, and the geometric drop profile was analyzed to calculate the droplet volume and interfacial area, and fitted with the Laplace-Young equation to determine the interfacial tension.The droplet volume was held constant by automatic regulation of the syringe.

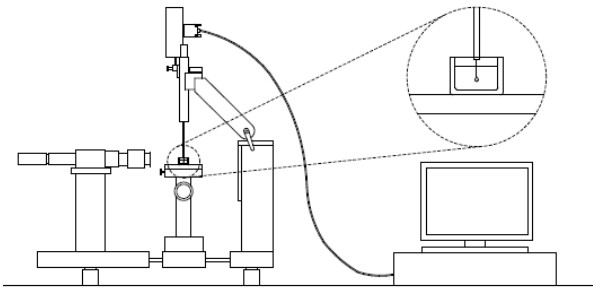

**Figure 1.** Pendant drop experimental setup.

### 3.2. Experimental Result: Effect of Enzyme Concentration

The decrease of interfacial tension between the two phases indicates the adsorption of interfacial active materials, such as enzymes, at the interface [32]. In Figure 2a the dynamic interfacial tension between sunflower oil and water is shown for different enzyme concentrations. In its native state, sunflower oil contains impurities (MG, DG, FA, waxes) which are surface active [5,42], thus decreasing the interfacial tension to 16 mN m$^{-1}$. The crude sunflower oil can be purified with column chromatography using Florisil® as adsorbent for separation of lipids [43]. The resulting interfacial tension of pure oil is 25.7 mN m$^{-1}$.

While there was an apparent influence of the impurities, it is believed that the adsorption of enzymes on the interface was more predominant compared to the other smaller molecules. This hypothesis was proven in another experiment using an oscillating drop, where the drop surface area was decreased and increased periodically by the syringe pump. Surfaces loaded with large molecules loose their elasticity [44], and have different dilatational modulus compared to ones loaded with small molecules [36,45].

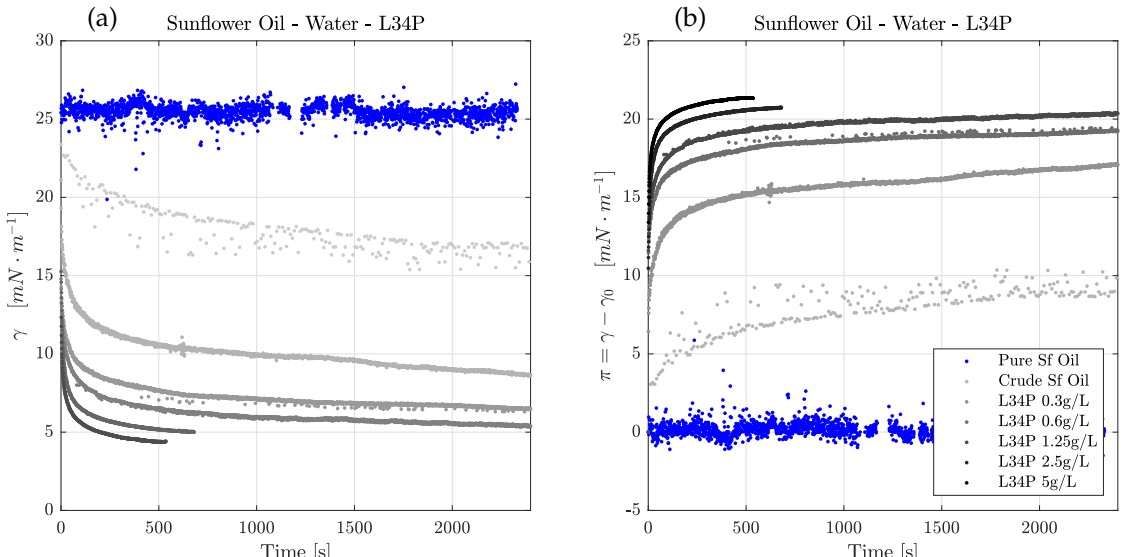

**Figure 2.** (**a**) Dynamic interfacial tension of sunflower oil and water at different enzyme (L34P) concentrations and (**b**) its reduction ($\pi = \gamma - \gamma_0$) due to adsorption of enzyme molecules.

With increasing enzyme concentration, the interfacial tension decreases rapidly, as seen in Figure 2. There were no sign of enzyme transfer from aqueous phase to the oil phase, which are usually indicated by increasing interfacial tension. Theoretically, when the interface is covered by a monolayer of surface active molecules, the adsorption equilibrium is reached, which is indicated by a constant interfacial tension through time. However, with large molecules such as protein and enzymes, the equilibrium

state is not fast to achieve. For one, diffusion of these molecules is slower than small molecules such as surfactants. This phenomenon causes the time dynamics of interfacial tension.

Secondly, large molecules possess a certain degree of flexibility, which can lead to molecular unfolding due to external forces, such as interfacial tension [37,38,46]. This can be seen at the lowest enzyme concentration (Ce = 0.3 g L$^{-1}$). The interfacial tension slowly decreased over time, and the equilibrium state was not reached after 48 h. After the initial decrease within the first minutes, the interfacial tension continued to decrease gradually, and after 96 h, the interfacial tension reached 6.4 mN m$^{-1}$, which is comparable to experiments with higher enzyme concentrations. This tendency confirms the finding of Miller et al. [35], that given enough area, enzyme molecules unfold at the interface in considerably long time. Similar phenomena were observed with different proteins [32,34,46,47], both for inert proteins and enzyme molecules, where the change of interfacial tension continues up to several days. It was concluded that the protein adsorption process at an interface is a multi-step process, including (i) diffusion of proteins from bulk phase towards interface or the induction phase, (ii) adsorption at the interface, (iii) change of molecular structure and (iv) spreading on the interface [32,46]. This leads to difficulties in the determination of the equilibrium point, as there is no definite point where adsorption is finished and the interfacial tension decrease is caused by molecular unfolding.

At higher concentration, the induction period diminished as the molecular diffusion is faster [26]. Looking at the dynamic interfacial tension in Figure 2a, it can be seen that experiments with enzyme concentration above 0.6 g L$^{-1}$ showed much faster adsorption rate, and the final interfacial tension ($\gamma$) can be achieved below 30 min. In Figure 2b, the reduction of interfacial tension is shown. It can be seen that the reduction are similar for high enzyme concentration. In Figure 3, the "steady state" interfacial tension at 30 min are plotted against the enzyme concentration (this time frame was chosen as the threshold, as the initial reaction rate in Section 4.5 was also calculated within the first 30 min. It is assumed that enzyme adsorption above this point does not affect the initial reaction rate). A saturation point is reached at 0.6 g L$^{-1}$ enzyme, where addition of enzymes after the saturation point does not reduce the interfacial tension significantly. It is assumed that above this point, the saturation coverage ($\Gamma_\infty$) was reached, as the interface is completely covered by enzyme molecules. This point marks the over-saturation of enzyme, as the additional enzymes do not have access to the substrates in the oil phase and therefore do not increase the reaction rate, which will be further demonstrated in Section 4.5.

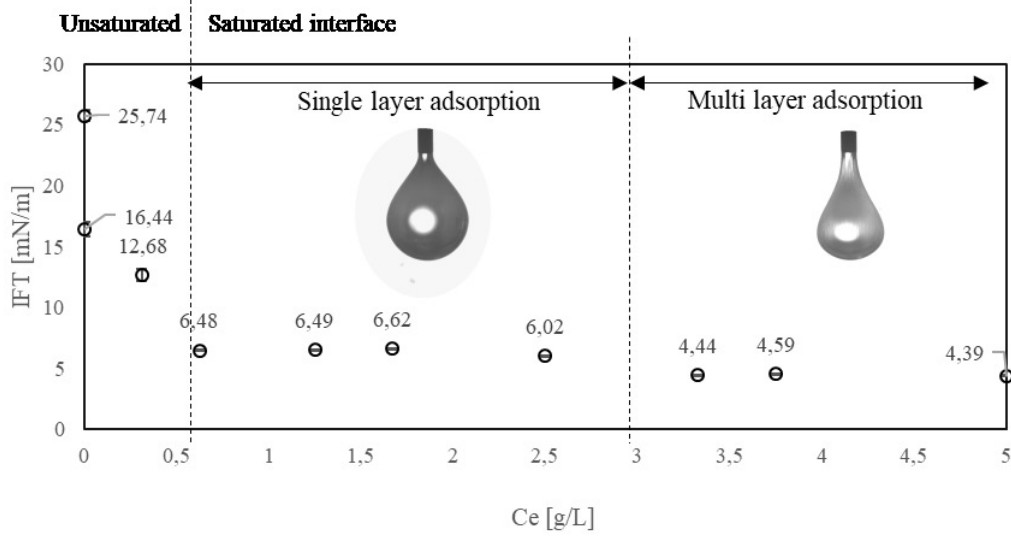

**Figure 3.** Interfacial tension of sunflower oil and water at different enzyme concentrations after 30 min. Error bar indicates the differences between multiple experimental data.

A small decrease of the interfacial tension was observed above Ce = 3 g L$^{-1}$, caused by a measurement error as multi-layer adsorption of the enzyme molecule appears on the interface, as shown in Figure 3. The deformed drop causes an error in the drop profile analysis [48]. This phenomenon is commonly observed with lipases and many other proteins. Large molecules are known to form a "skin" at interfaces [28,32], which also promotes necking and detachment of the drops. It has been attributed to protein unfolding [36] and is usually pronounced by the formation of inter-molecular bonds with phospholipid [28] or other large molecules which stabilizes the interface [49]. Oscillating drop experiments have the potential to determine the extent of reversibility of adsorption at this state [50]. However, more in depth experiments are needed.

### 3.3. Modeling of Enzyme Adsorption Kinetics

In this work, the interfacial enzyme loading ($\Gamma_\infty$) and the equilibrium adsorption constant ($K_L$) were calculated by fitting adsorption isotherm against experimental data using Gibbs' fundamental isotherm (Equation (4)) and the Langmuir adsorption isotherm (Equation (5)). The initial interfacial tension ($\gamma_0$) was assumed as a constant from the interfacial tension of pure oil and water, where the enzyme molecules were diluted in [51].

$$\gamma = \gamma_0 + RT\Gamma_\infty ln\left(1 - \frac{\Gamma}{\Gamma_\infty}\right) \tag{4}$$

$$\Gamma = \Gamma_\infty \frac{K_L \cdot E}{1 + K_L \cdot E}; \quad K_L = \frac{k_a}{k_d}. \tag{5}$$

It is to be noted that the Langmuir isotherms made the following assumptions:

- Enzymes are adsorbed in single layer configuration.
- One molecule occupies one adsorption site and is assumed to be globular without molecular unfolding.
- No adsorption competition from other interfacial active components such as DG, MG and FA.

### 3.4. Modeling Result: Enzyme Adsorption Kinetics

The best parameter fit is shown in Table 1. Figure 4 shows good agreement between simulation and experimental data, which provides the Langmuir constant $K_L$ and the maximum interfacial loading $\Gamma_\infty$ of the enzyme. However, due to complex adsorption behaviour of large molecules (slow diffusion to the interface, molecular unfolding and the multi layer adsorption at the w/o interface) it is difficult to calculate these values accurately. Latour [27] demonstrated that fitting protein adsorption to Langmuir isotherm cannot explain the real behaviour, especially due to (1) molecular unfolding on interface, (2) multi layer adsorption and (3) interaction between molecules.

As mentioned before in the pendant drop experiments, at low concentration (Ce = 0.3 g L$^{-1}$), the enzyme molecules spread, occupy larger area, while at higher concentration (Ce = 2.5 g L$^{-1}$), the molecules are tightly packed, and possibly form multi layer adsorption. These facts should to be taken into account while determining the $\Gamma_\infty$ and $K_L$. According to Latour [27], the value of $K_L$ should be adapted for different ranges of protein concentrations, in order to describe the molecular spreading and the change of adsorption process due to molecule interaction.

Furthermore, empirical correlations such as the Gibbs' fundamental equation or the Langmuir adsorption isotherm are derived for small molecules, where the diffusion process is much faster compared to protein molecules [52]. These isotherms cannot predict the time dynamic interfacial tension as shown in Figure 3. The equation proposed by Ward and Tordai [53] models the time evolution of interfacial loading ($\Gamma$) for diffusion controlled adsorption. The model describes molecular transfer by diffusion from the bulk phase to a "subsurface" in the vicinity of the w/o interface, where the molecules can then be adsorbed to the interface. Stasiak et al. [54] demonstrated that

the Ward Tordai equation can be used to determine the apparent diffusion coefficient depending on the bulk concentration. For systems with slow molecular diffusion (for example with large molecules as catalysts), the apparent diffusion coefficient can indicate whether a system exhibits mass transfer limitation.

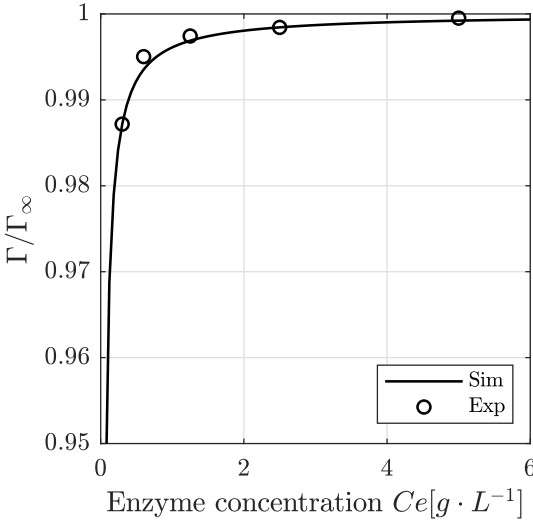

**Figure 4.** Fitting of interfacial loading at steady state for different enzyme concentrations.

**Table 1.** Best fit adsorption kinetics, simulation result shown in Figure 4.

| Parameter | | Value | Unit |
|---|---|---|---|
| Langmuir constant | $K_L$ | 10021 | $\mathrm{m^3\,mol^{-1}}$ |
| Max enzyme loading | $\Gamma_\infty$ | $1.65 \times 10^{-6}$ | $\mathrm{mol\,m^{-2}}$ |
| Pure IFT | $\gamma_0$ | 25.74 | $\mathrm{mN\,m^{-1}}$ |
| Error | SSE $\Gamma$ | $3.84 \times 10^{-4}$ | $\mathrm{mol\,m^{-2}}$ |

To address the multi layer adsorption, Maldonado-Valderrama et al. [34] modelled the protein adsorption process based on a theory that multiple states of the protein molecule can exist at the interfacial layer. However, the application of the model is difficult due to the lack of data, as the model requires the molar area of the protein molecule at the different states.

Despite all the simplification, the isotherms were able to quantify the maximum interfacial loading $\Gamma_\infty$ and the Langmuir constant $K_L$ of the enzyme adsorption process. These values represented the adsorption process in a single drop, and were used to calculate the maximum interfacial coverage ($A_t$, Equation (14)) in the emulsion and as initial value for the adsorption and desorption constant ($k_a/k_d$) for fitting the reaction constants (Equation (16)). More complicated models, as mentioned above, can be used to predict the values more accurately, but deemed to be unnecessary for the reaction kinetic modelling, which is the main scope of this work.

## 4. Hydrolysis Reaction in Emulsion Reactor

The reaction kinetic constants were fitted against experiment data from hydrolysis reactions in an emulsion reactor. Experimental conditions were designed to demonstrate the reaction limitation by surface area at specific enzyme concentrations. This was done by performing the reaction in water-in-oil emulsions, where the amount of enzyme per gram substrate can be kept at a minimum. The water to oil volume ratio was kept constant as 3 to 10. With this ratio, the amount of substrates (water ($H_2O$) and glyceride ($G$)) molecules were always in excess for all experiments. Therefore, the reaction rates were not dependent on the concentration of the substrates, but on the specific interfacial area ($a$) and the enzyme concentrations ($Ce$).

### 4.1. Experimental Methodology

#### 4.1.1. Batch Reaction in STR

Batch experiments were conducted in a stirred tank reactor with a total working volume of 13 L. All reactions were done at 30 °C, the reactor was equipped with a double wall for temperature control. The detailed dimensions are shown in Figure 5. Three Rushton turbines were distributed evenly along the height of the reactor, with three baffles near the reactor wall. The complete system had a total power number ($Ne$) of 9.5. The power number for the specific geometry was experimentally determined by torque measurements (Equation (6)). By using a slim reactor with multistage impellers, it was possible to operate at low shear to create emulsions with larger droplets while ensuring homogeneous mixing, in order to demonstrate the limitation of interfacial area to reaction rate. Several batch experiments (Table 2) were conducted to evaluate the effect and interaction of power input ($P/V$) and the enzyme concentration ($Ce$) on the drop size and subsequently the reaction rate.

$$P = Ne \cdot \rho \cdot n^3 \cdot D^5 = M \cdot \omega. \tag{6}$$

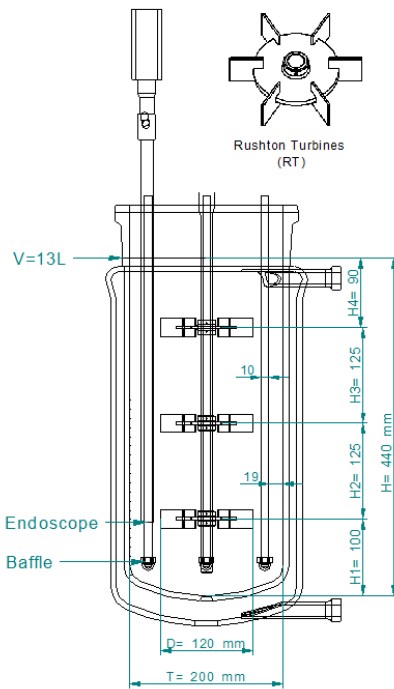

**Figure 5.** Reactor setup with in-situ drop size measurements.

**Table 2.** Experiment matrix and their discrepancies from modeling results.

| Exp No. | Stirring Speed $n$ $s^{-1}$ | Power Input $P/V$ $W\,m^{-3}$ | Enzyme Concentration $Ce$ $g\,L^{-1}$ | $Et$ $mol\,L^{-1}$ | Drop Diameter $d_{32}$ $\mu m$ | Adsorption Site $A_t$ $mol\,L^{-1}$ | Error SSE $(FA+TG)$ $mol\,L^{-1}$ |
|---|---|---|---|---|---|---|---|
| 1 | 4.17 | 1300 | 1.25 | $7.21 \times 10^{-6}$ | 61.85 | $4.14 \times 10^{-5}$ | 0.0586 |
| 2 | 2.5 | 300 | 1.25 | $7.21 \times 10^{-6}$ | 115.64 | $1.93 \times 10^{-5}$ | 0.0419 |
| 3 | 1.67 | 80 | 1.25 | $7.21 \times 10^{-6}$ | 199.16 | $8.25 \times 10^{-6}$ | 0.0555 |
| 4 | 2.5 | 300 | 0.6 | $3.46 \times 10^{-6}$ | 162.81 | $1.45 \times 10^{-5}$ | 0.0590 |
| 5 | 1.67 | 80 | 0.6 | $3.46 \times 10^{-6}$ | 340.82 | $2.99 \times 10^{-6}$ | 0.0675 |
| 6 | 2.5 | 300 | 2.5 | $1.44 \times 10^{-5}$ | 100.59 | $2.34 \times 10^{-5}$ | 0.0794 |
| 7 | 1.67 | 80 | 2.5 | $1.44 \times 10^{-5}$ | 165.05 | $1.42 \times 10^{-5}$ | 0.0990 |

### 4.1.2. Analysis Methods

Drop Size Distribution

In the experiments, the droplet size was influenced by varying the specific power input ($P/V$). The drop sizes are measured in-situ in the reactor using an endoscope probe, based on Maaß et al. [55]. An algorithm was developed to recognize the droplets automatically, as seen in Figure 6b. The algorithm gives statistical data representing the drop size distribution. Sauter mean diameter ($d_{32}$) was chosen as the representative drop size, as it reflects the mean ratio between the volume and interfacial area of the drops (Equation (7)). The drop sizes depend on the stirring speed or the power input, as well as the interfacial tension, as shown in Equation (10).

$$d_{32} = \frac{\sum\limits_{i=1}^{n} d_i^3}{\sum\limits_{i=1}^{n} d_i^2} \tag{7}$$

$$\frac{d_{32}}{d} = C \cdot We^{-0.6}(1 + b\,\phi_{vd}) \tag{8}$$

$$We = \frac{\rho_c n^2 D^3}{\gamma} \tag{9}$$

$$\phi_{vd} = \frac{V_{aq}}{V_t}. \tag{10}$$

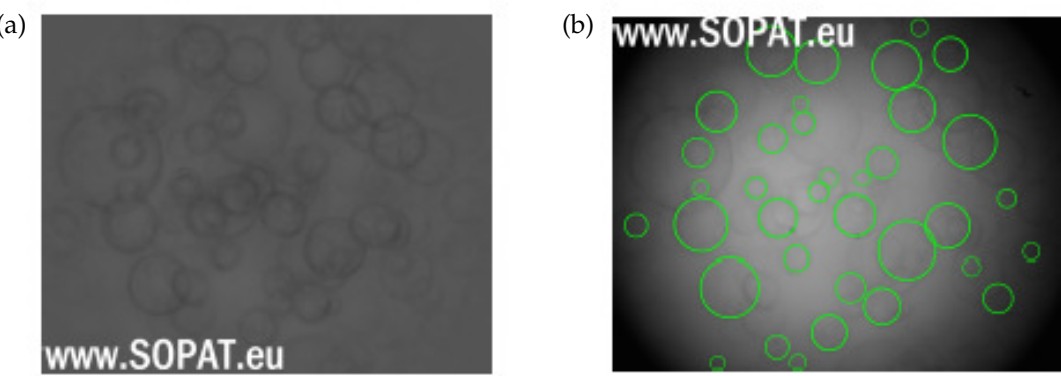

**Figure 6.** (**a**) Water drops in sunflower oil, (**b**) with drop recognition algorithm.

Concentration Analysis

High pressure size exclusion chromatography (HPSEC) was conducted for composition analysis of tri- , di-, monoglyceride and fatty acid molecules (TG, DG, MG, and FA) based on their molecular size [56]. 25 μg of oil samples were diluted in 5 mL of THF. 20 μL of the diluted sample was injected manually with a sample loop into the HPLC system (AZURA® Analytical HPLC, Knauer, Berlin, Germany). Separation was done in PSS SDV Analytical 50 Å Column (300 × 8 mm) with particle size of 5 μm (PSS Polymer Standards Service GmbH, Mainz, Germany) at 40 °C. THF was used as eluent, with modification of 0.3% acetic acid at flowrate of 1 mL min$^{-1}$. Refractive Index Detector (RID) was used to detect the signals.

Determination of Reaction Rate and Hydrolysis Degree

Samples were taken from the reaction at certain time steps up to 6 h to determine the two important parameters in the reaction: the initial reaction rate ($r_{FA,0}$) and the equilibrium hydrolysis degree (*DH*).

Initial reaction rate ($r_{FA,0}$) was calculated by the slope of the first 30 min of reaction by linear regression with MATLAB® of the experimental points. For experiments with higher reaction rate (high enzyme concentration and high rotational speed), the initial reaction rate was evaluated for smaller time intervals (2–4 min), until the linear R-squared (goodness-of-fit measure) is less than 0.9. It was assumed that the reaction started when mixing started, as without sufficient surface area, the reaction rate can be negligible.

$$r_{FA,0} = \frac{1}{V_R} \frac{n_{FA}(t) - n_{FA,0}}{t}. \tag{11}$$

The hydrolysis degree (DH) is defined as the amount of fatty acid (*FA*) produced at a certain time, compared to the total amount of FA possible in the initial sample.

$$DH(t) = \frac{n_{FA}(t)}{3 \cdot n_{TG,0} + 2 \cdot n_{DG,0} + n_{MG,0} + n_{FA,0}}. \tag{12}$$

### 4.2. Experimental Result: Emulsification and Drop Size Distribution

The effect of enzyme adsorption at the w/o interface can also be seen in the drop swarm/emulsion. Figure 7 shows the drop size in emulsions with lipase concentrations between 0.6–2.5 g L$^{-1}$. At these concentrations, the interfacial tensions measured in Section 3.1 are similar, ranging between 6.0–6.48 mN m$^{-1}$ (see Figure 3), which should theoretically lead to similar drop sizes. The actual drops were smaller in emulsions with higher enzyme concentrations. It indicates that the adsorption of lipases at the interface hinders drop coalescence. The effect is further promoted at higher enzyme concentration, as more interfacial area is created with smaller drops and more enzyme molecules are adsorbed at the interface. At higher stirring speed (high $P/V$), there were little influence of enzyme concentration, as droplet breakup is more dominant than coalescence (see Figure 7b). The drop size determines the available interfacial area in the reactor (Equation (14)), thus determines the reaction rate.

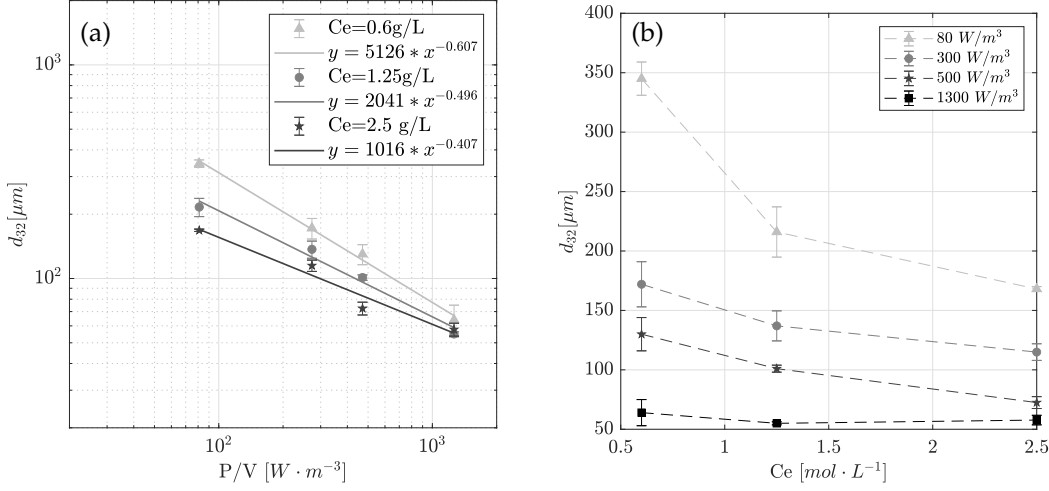

**Figure 7.** Drop sizes in sunflower oil—water emulsions with (**a**) different L34P concentration and (**b**) power input.

Figure 8 shows representative drop size distributions for the different enzyme concentration at the same power input ($P/V$ = 80 W m$^{-3}$). The number based distribution ($Q_0$) shows that most of the drops were smaller than 100 µm, with only few large drops. This confirms the hypothesis

that the system was dominated by breakup while coalescence was hindered, especially with higher enzyme concentration.

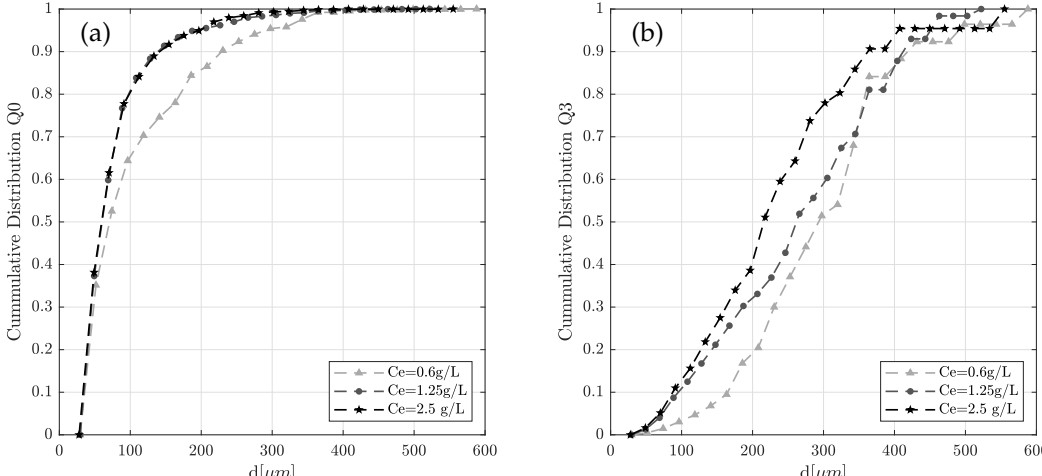

**Figure 8.** (**a**) Number based and (**b**) volume based cumulative drop size distribution in sunflower oil—water emulsions with different L34P concentration and power input.

### 4.3. Modeling of Enzymatic Reaction Kinetics

To model the enzymatic reaction, the reversible Michaelis Menten (MM) kinetics based on Keleti [18] was chosen due to its simplicity. Several studies [11,13,14,16] demonstrated that the hydrolysis reaction can be modeled using the one step reaction kinetics (Equation (3)), as long as there are no significant effects from the side products. In this model, there is no differentiation between tri-, di- and monoglyceride (TG/DG/MG) as substrates. Each molecule is assumed to contain three, two and one molecules of glyceride respectively. This assumption was made as L34P has no specificity towards the glyceride ligands. The total glyceride ($G$) was calculated as shown in Equation (13). The concentration of reaction species are expressed per reactor volume ($V_R$).

$$n_G = 3 \cdot n_{TG} + 2 \cdot n_{DG} + n_{MG}$$
$$G = \frac{n_G}{V_R}. \tag{13}$$

To be integrated in the model, the specific interfacial area ($a$) is expressed as the amount of enzyme molecules that can be adsorbed ($A_t$). The maximum enzyme interfacial loading ($\Gamma_\infty$) was taken from experimental fitting in Section 3.3. $A_t$ represents the total number of molecules which can be adsorbed at the interface, consisting of the free adsorption sites $A$ and the area occupied by enzyme molecules $E^*$ and enzyme complexes $EG$.

$$A_t = a \cdot \Gamma_\infty = \frac{6 \cdot V_{aq}}{d_{32}} \cdot \frac{1}{V_R} \cdot \Gamma_\infty$$
$$A = A_t - E^* - EG. \tag{14}$$

The total enzyme concentration $E_t$ is calculated from the enzyme concentration $Ce$ and its molecular mass $Mr$. Enzyme molecules are divided into enzymes in the aqueous phase $E$, enzymes on the interface $E^*$ and enzyme complexes $EG$.

$$E_t = Ce \cdot Mr \cdot \frac{V_{aq}}{V_R}$$
$$E = E_t - E^* - EG. \tag{15}$$

The reaction kinetics were integrated with the following ODEs.

$$\frac{dG}{dt} = k_{-1} \cdot EG - k_1 \cdot E^* \cdot G$$

$$\frac{dFA}{dt} = k_2 \cdot EG - k_{-2} \cdot FA \cdot E^*$$

$$\frac{dEG}{dt} = k_1 \cdot E^* \cdot G - k_{-1} \cdot EG - k_2 \cdot EG + k_{-2} \cdot FA \cdot E^*$$

$$\frac{dE^*}{dt} = k_a \cdot E \cdot a - k_d \cdot E^* - k_1 \cdot E^* \cdot G + k_{-1} \cdot EG + k_2 \cdot EG - k_{-2} \cdot FA \cdot E^*.$$

(16)

Global fitting of the six reaction parameters was done with MATLAB $^®$ 2019b against *FA* and *G* concentration in experiment 1–7 (see Table 2). As mentioned before, fitting multiple parameters simultaneously can result in arbitrary results. It was not possible to fit one or two constants separately from the other constants, as the co-linearity between the constants are high. Therefore, to simplify the fitting, a systematic fitting was implemented. Multiple experiments were fitted to avoid ad-hoc result, where the rate constants are only applicable to a certain condition. First of all, experiments 1, 4 and 6 were used for fitting reaction constants $k_1$ and $k_2$. These experiments were done at high stirring speed, where no surface limitation should occur and the effect of adsorption rates can be neglected. Afterwards, experiment 3, 5 and 7 were fitted to determine $k_a$ and $k_d$. As initial value, the Langmuir constant fitted in Section 3.3 ($K_L = k_a / k_d = 1.00 \cdot 10^4$ m$^3$ mol$^{-1}$) was used. The final parameters used in this work are listed in Table 3.

**Table 3.** Enzymatic reaction kinetic constants. Result from experimental fitting, except *Mr* (from literature) and $\Gamma_\infty$ (from Section 3.3).

| Parameter | | Value | Unit |
|---|---|---|---|
| Molecular mass | *Mr* | 40,000 | Da |
| Enzyme loading | $\Gamma_\infty$ | $1.65 \times 10^{-6}$ | m$^3$ mol$^{-1}$ |
| Adsorption rate constant | $k_a$ | $1.86 \times 10^{11}$ | L mol$^{-1}$ min$^{-1}$ |
| Desorption rate constant | $k_d$ | $4.43 \times 10^4$ | min$^{-1}$ |
| Forward reaction rate constant | $k_1$ | $1.97 \times 10^3$ | L mol$^{-1}$ min$^{-1}$ |
| Backward reaction rate constant | $k_{-1}$ | $3.50 \times 10^3$ | min$^{-1}$ |
| Forward reaction rate constant | $k_2$ | $2.36 \times 10^5$ | min$^{-1}$ |
| Backward reaction rate constant | $k_{-2}$ | $6.73 \times 10^3$ | L mol$^{-1}$ min$^{-1}$ |
| Langmuir constant | $K_{L,V_R}$ | $4.20 \times 10^7$ | L mol$^{-1}$ |
| Substrate Michaelis Menten constant | $K_{M,S}$ | 121.26 | mol L$^{-1}$ |
| Product Michaelis Menten constant | $K_{M,P}$ | 35.61 | mol L$^{-1}$ |

## 4.4. Reaction Time Course

A typical reaction course can be seen in Figure 9. Sunflower oil splitting with L34P typically reaches a hydrolysis degree (*DH*) up to 93% within the first 6 h of a batch experiment. There were no accumulation of side products (*DG* or *MG*). However, complete hydrolysis were not reached, leaving 7% of unconverted *TG*. Lipases are known to catalyze the backwards reaction, converting *FA* and *Gly* into *MG* or *DG* molecules. This tendency depends on the affinity of the specific enzyme towards the product or the substrate molecules. It is more dominant in transesterification reactions for biodiesel production [3,4,57–59] compared to hydrolysis reactions [15,16], but the effect cannot be neglected in a continuous process. Therefore, to model this system, the reversible Michaelis-Menten mechanism (Equation (3)) was chosen. Keleti [18] defined the Michaelis Menten constants as shown in Equation (17) and (18). These constants give indication of enzyme specificity towards the product or the substrate. Higher $K_{M,P}$ indicates higher backwards reaction or lower hydrolysis degree (*DH*).

In this system, $K_{M,S}$ is much higher than $K_{M,P}$ (see Table 3), which represents the high final hydrolysis degree (*DH*) of 93 %.

$$K_{M,S} = \frac{k_{-1} + k_2}{k_1} \tag{17}$$

$$K_{M,P} = \frac{k_{-1} + k_2}{k_{-2}}. \tag{18}$$

The lines in Figure 9 represent the simulation results. The model can predict not only the initial rate of reaction, as well as the final equilibrium hydrolysis. The model shows good agreement with the experimental data for *FA* production and *G* consumption (Figure 10), although with the one step kinetics, only the consumption of the main substrate (*G*, representing all glyceride molecules) and the production of the main product (*FA*) can be modelled. The influence of water ($H_2O$) and glycerol (*Gly*) as second substrate and side product, as well as the Di- and Mono- glycerides (*DG*, *MG*) as intermediate products are not taken into account. This assumption was made, as the aim of the model is only to demonstrate the limitation of interfacial area. For systems with high inhibitory effect, such as the splitting of castor oil or the transesterification reaction to produce biodiesel, Ping-Pong Bi-Bi kinetics, which includes two substrates in the reaction, are needed [3,58].

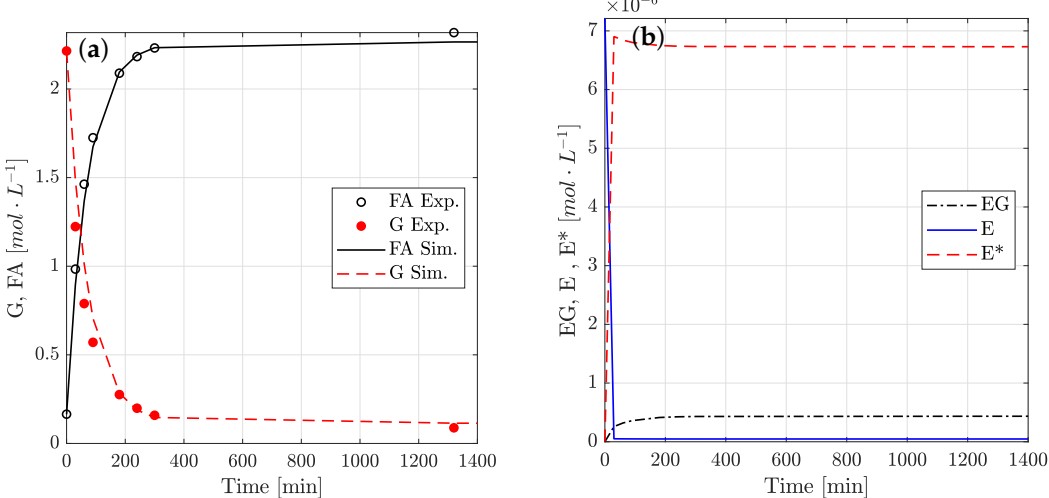

**Figure 9.** Time dynamics of (**a**) product and substrate and (**b**) enzyme complex concentrations, simulated and experimental data from experiment 1. Rate constants are listed in Table 3.

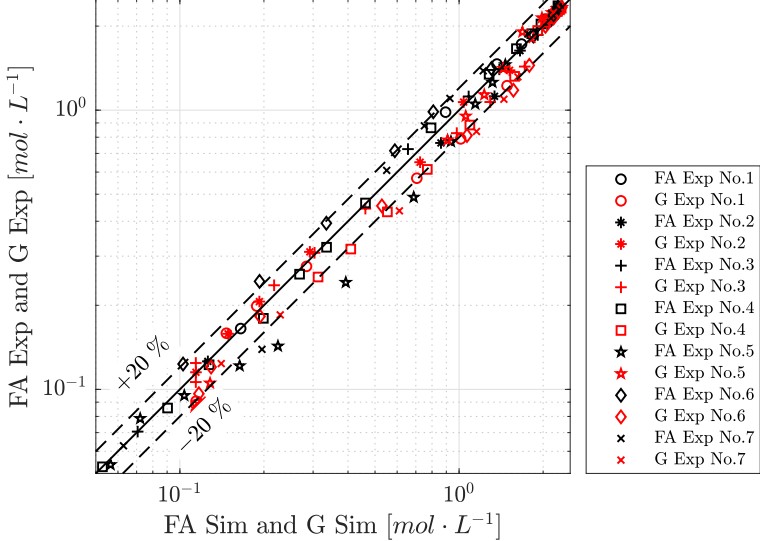

**Figure 10.** Parity plot, triglyceride consumption and fatty acid production.

### 4.5. Limitation of Interfacial Area to Reaction Rate

Seven batch experimental results were chosen for discussion in this paper, as listed in Table 2, with varying power input and enzyme concentration to influence the drop size. In Figure 11a, the initial reaction rates measured and calculated in Section 4 are plotted against the interfacial area. The graph shows two tendencies: at low interfacial area with large droplets, the reaction rate is limited by the interfacial area. When the power input is increased, the drops are smaller and the interfacial area ($a$) is larger. At one point, there is sufficient interfacial area for all enzyme molecules to be adsorbed. Thereafter, the reaction rate is not limited by the interfacial area ($a$), but only by the enzyme concentration ($Ce$, subsequently $E^*$).

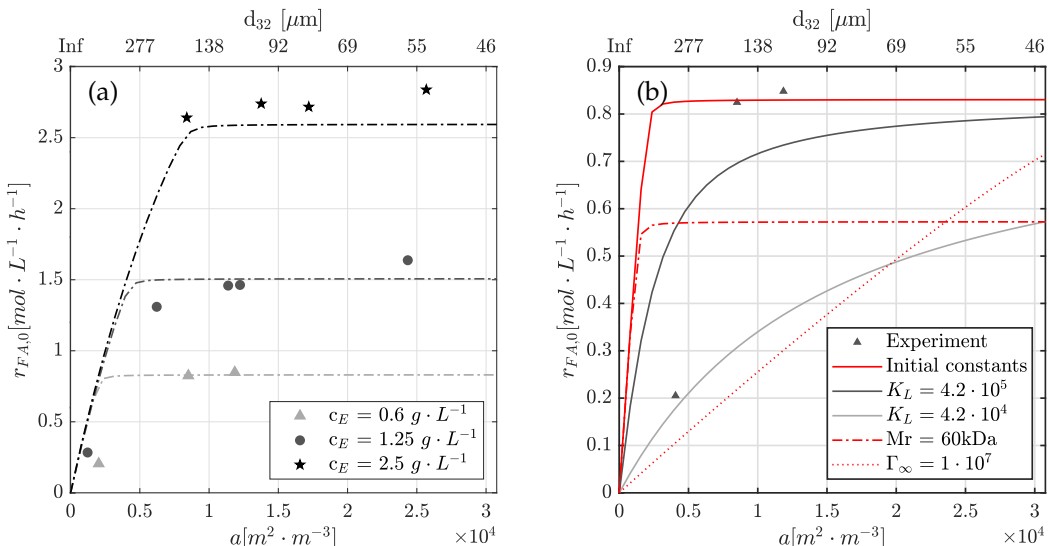

**Figure 11.** (**a**) Dependence of reaction rate on interfacial area, and (**b**) the dependence of the saturation point to the adsorption constant ($K_L$), the maximum enzyme loading ($\Gamma_\infty$) and the molecular weight ($Mr$). Initial constants are listed in Table 3.

The simulation, represented as the dotted lines in Figure 11a, is able to predict the limitation of surface area and determine the minimum interfacial area ($a$), or the maximum drop size ($d_{32}$) in the emulsion reactor at a certain enzyme concentration. This is the optimum condition for a stirred tank reactor, where the reaction is at maximum and the shearing is kept minimum to avoid enzyme deactivation.

In order to determine this optimum point, the values of the enzyme loading at the interface $\Gamma_\infty$ (Equation (14)) and the molecular mass $Mr$ of the enzyme (Equation (15)) are needed. Both $\Gamma_\infty$ and $Mr$ influence the saturation point, as shown in Figure 11b. As $\Gamma_\infty$ was fitted in Section 3.3 with $Mr = 40$ kDa, the model predicts that enzymes with higher molecular weight (for example $Mr = 60$ kDa) occupy larger area on the interface and less enzymes are adsorbed on the interface (lower $E^*$). Therefore, the maximum reaction rate is reduced. $\Gamma_\infty$ influences the dependence of reaction rate to the specific interfacial area area ($a$). With lower $\Gamma_\infty$, less enzymes can be adsorbed per area, which also leads to slower reaction rates. The data from the single drop modeling (Section 3.1) are necessary to predict the proper enzyme loading ($\Gamma_\infty$) with a specific molecular mass.

### 4.6. Physical Significance of Rate Constants

While the enzymatic hydrolysis kinetics have been extensively studied, it is difficult to find agreement in literature about the value of reaction constants. These parameters depend on the combination of enzyme and substrate, as well as the process conditions (temperature, phase ratio, etc.). Furthermore, as commonly encountered with numerical fitting of complex kinetics, more than

one combination of constants can be found to reach the optimum fit. While fitting the constants to experimental data, the absolute values of the individual rate constants are dependent on the lower and upper boundary of the solver, and thus can have arbitrary values without a real physical meaning. The plausibility of the set of constants must be analyzed with physical explanations.

Although the absolute values of the constants can be arbitrary, the ratio between the constants gives an indication of the physical phenomena. The value of $K_L$ in comparison to $K_{M,S}$ is important in the macrokinetics, as it indicates whether the system displays mass transfer limitation of the enzyme transfer to the w/o interface. The influence of $K_L$ is shown in Figure 11b. At the same specific interfacial area, low $K_L$ (low $k_a$ or high $k_d$) leads to slower reaction, as not all enzymes were adsorbed on the interface ($E^* \ll E$). In this particular system, as indicated in Section 3, the adsorption rate is much faster than the reaction rate. Although the catalysts are large molecules, whose adsorption are influenced by diffusion, the enzyme interfacial adsorption takes place within seconds, while the first *FA* concentration can be detected within minutes from the start of reaction. This is represented with high value of $k_a$ compared to other constants. At these rates, the maximum enzyme concentration at the interface $E^*$ were reached, and the limiting factor was the reaction rate itself.

Furthermore, the concentration of enzyme complexes ($EG$, $E$ and $E^*$) in Figure 9b are good indicators for the plausibility of reaction constants. The high $K_a$ caused the steep increase in the enzyme concentration on the interface ($E^*$) and the fast decrease of enzymes on the bulk phase ($E$). As the reaction progressed, the enzyme complex $EG$ is produced, but as the molecules are thermodynamically unstable, they rapidly split into *FA*, releasing $E^*$ for further reaction. After the initial reaction rate, when substrate concentration $G$ is decreasing, $E^*$ does not rapidly turn into $EG$. At this state, an equilibrium between $E^*$ and $EG$ is reached. Therefore, no high concentration of $EG$ should be present in the system. This indicates inappropriate fitting of $k_{-1}$ and $k_2$ compared to $k_1$ and $k_{-2}$.

## 5. Conclusions

In this paper, the importance of the interfacial area and enzyme adsorption for the hydrolysis reaction with lipases is demonstrated, both experimentally and numerically. The system showed a heterogeneous catalytic tendency, where the reaction rate depends strongly on the availability of w/o interfacial area. The saturation coverage is dependent on the enzyme concentration ($Ce$), the molecular size ($Mr$) and the maximum enzyme interfacial coverage $\Gamma_\infty$. By including an adsorption term to the basic one-step enzymatic reaction mechanism, the model can predict the limitation of interfacial area. The maximum enzyme interfacial coverage $\Gamma_\infty$ was fitted from enzyme adsorption data in a single drop with a specific interfacial area ($a$). With this value, the saturation point in an emulsion reactor can be calculated and the optimum operational condition in regards of total surface area or the stirring speed can be determined.

Furthermore, by simplifying the system from drop swarm to a single drop, it was possible to observe the adsorption rate of the protein molecules. It was clear that the bulk enzyme concentration affects the apparent rate of adsorption. The higher the bulk concentration, the faster the decrease of interfacial tension. This phenomenon can also be attributed to the faster molecular diffusion to the interface, or due to change of adsorption process because of molecular interaction. Further modelling is needed to differentiate the different effects, although for the purpose of modelling the limitation of interfacial area, fitting with Langmuir isotherm was sufficient.

By combining the experimental data and numerical investigation in two different scales (single drop and drop swarms), the developed model is able to predict the optimum interfacial area or drop size in an emulsion reactor for the hydrolysis reaction. However, since a single step (Michaelis Menten) kinetics was chosen, the model cannot predict the inhibitory effects of the second substrate nor intermediates and side products. Further modeling using Ping-Pong Bi-Bi kinetics has the potential to solve this problem, providing that the complexity of the parameter fitting can be reduced.

**Author Contributions:** S.R. conceived and designed the experiments and wrote the paper; J.G. wrote the numerical code and performed the experiments. A.D. and M.K. supervised. All authors have read and agreed to the published version of the manuscript.

**Funding:** This project has received funding from the Bio Based Industries Joint Undertaking (BBI JU) under the European Union's Horizon 2020 research and innovation program under grant agreement No. 720743. https://www.bbi-europe.eu/projects/lipes.

**Conflicts of Interest:** The authors declare no conflict of interest. The founding sponsors had no role in the design of the study; in the collection, analyses, or interpretation of data; in the writing of the manuscript, and in the decision to publish the results.

## Abbreviations

The following abbreviations are used in this manuscript:

| | |
|---|---|
| DG | Diglyceride |
| DH | Degree of Hydrolysis |
| FA | Fatty Acid |
| G | Glyceride moiety |
| Gly | Glycerol |
| HPLC | High Performance Liquid Chromatography |
| L34P | Lipase enzyme Lipomod 34P |
| MG | Monoglyceride |
| MM | Michaelis Menten |
| ODE | Ordinary Differential Equation |
| PPBB | Ping Pong Bi Bi |
| TG | Triglyceride |
| SSE | Sum of Square Error |

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
