# Peer review of "A Multi-Scale Approach to Modeling the Interfacial Reaction Kinetics of Lipases with Emphasis on Enzyme Adsorption at Water-Oil Interfaces"

_processes, doi:10.3390/pr8091082_

Round 1
Reviewer 1 Report
The article entitled “A multi-scale approach to modeling the interfacial reaction kinetics of lipases with emphasis on enzyme adsorption at water-oil interfaces” describes the importance of the interfacial area and enzyme adsorption for the hydrolysis reaction with lipases is demonstrated, both experimentally and numerically. The most interesting aspect is combining the experimental data and numerical investigation in two different scales (single drop and drop swarms), the developed model is able to predict the optimum interfacial area or drop size in an emulsion reactor for the hydrolysis reaction.
The paper has minor revisions and therefore we recommend it is highly suitable for publication in Processes. The minor revisions are:
Title:
- Author should check the spelling correction for the title (interfacial reacion instead of interfacial reaction)
Materials and Methods:
- The author should add Sub subsection throughout the paper
- The author should mention short description of figure 1 & 2.
- In Figure 3 author should add the error bar

Author Response
The article entitled “A multi-scale approach to modeling the interfacial reaction kinetics of lipases with emphasis on enzyme adsorption at water-oil interfaces” describes the importance of the interfacial area and enzyme adsorption for the hydrolysis reaction with lipases is demonstrated, both experimentally and numerically. The most interesting aspect is combining the experimental data and numerical investigation in two different scales (single drop and drop swarms), the developed model is able to predict the optimum interfacial area or drop size in an emulsion reactor for the hydrolysis reaction.
Thank you for your comment.
The paper has minor revisions and therefore we recommend it is highly suitable for publication in Processes. The minor revisions are:
Title:
- Author should check the spelling correction for the title (interfacial reacion instead of interfacial reaction)
Done. Sorry for the mistake.
Materials and Methods:
- The author should add Sub subsection throughout the paper
The sections are now divided into more sub-sub-section. However, I would like to avoid too many levels (e.g. sub-sub-sub-section), especially when the content consist only one paragraph.
- The author should mention short description of figure 1 & 2.
Done. Description for Figure 2 is spread throughout Section 3.2.
- In Figure 3 author should add the error bar
Done. Only the error bars are very small, they are hidden behind the data points.

Reviewer 2 Report
Rusli and co-workers presented, both experimentally and numerically, the importance of the enzyme adsorption and interfacial area in lipase-based hydrolysis of sunflower oil. Despite the fact that the work is very interesting, the introduction provides a strong background and results of the work are valuable and show originality, the authors did not avoid some minor errors, which I will present below.
- The abstract is too long. According to "Instructions for Authors" (https://www.mdpi.com/journal/processes/instructions) - The abstract should be a total of about 200 words maximum.
- Equation 1 is not described in the text. Besides, are equation 1 and equation 4 necessary in the manuscript?
- In my humble opinion explanation of symbols, constants or abbreviations should be also provided in table footers. All explanations are given at the end of the manuscript, but when are listed next to the table, it is much easier to understand the data.
- Table captions should appear above the tables.
- In some cases, descriptions of experiments, methods used, and results obtained should be separated and transferred to the appropriate sections. For example, "Concentration analysis" and methodology for HPSEC should be transferred to "Materials and methods" section.
Author Response
Rusli and co-workers presented, both experimentally and numerically, the importance of the enzyme adsorption and interfacial area in lipase-based hydrolysis of sunflower oil. Despite the fact that the work is very interesting, the introduction provides a strong background and results of the work are valuable and show originality, the authors did not avoid some minor errors, which I will present below.
Thank you for your comments.
- The abstract is too long. According to "Instructions for Authors" (https://www.mdpi.com/journal/processes/instructions) - The abstract should be a total of about 200 words maximum.
The abstract is now shortened (194 words). Sorry for the mistake.
- Equation 1 is not described in the text. Besides, are equation 1 and equation 4 necessary in the manuscript?
Equation 1 is described in the second paragraph in the introduction, I moved the equation closer to the description. I think it's important to show that in reality two substrates are involved in the reaction, while the reaction kinetic model only take one substrate into account. For equation 4, I agree it is redundant.
- In my humble opinion explanation of symbols, constants or abbreviations should be also provided in table footers. All explanations are given at the end of the manuscript, but when are listed next to the table, it is much easier to understand the data.
I understand the problem. The symbols, constants and abbreviations in the tables are now attached with the description, not as table footers but included in the table itself.
- Table captions should appear above the tables.
Done. Sorry for the mistake.
- In some cases, descriptions of experiments, methods used, and results obtained should be separated and transferred to the appropriate sections. For example, "Concentration analysis" and methodology for HPSEC should be transferred to "Materials and methods" section.
This was done on purpose to separate the methods used in the different scales. The materials used in both scales were the same, but the analysis is different- HPSEC, for example, was only used in the emulsion reactor, not in the single drop experiments. Therefore only the general methodology (both experimental and numerical) is listed in Section 2, and in each scales (Section 3 and 4), the specific methodology is again listed. I altered the description of the subsections and the title, to make the distinction between experimental methodology and results and numerical.
